# Preparation and Performance Study of High-Strength and Corrosion-Resistant Cement-Based Materials Applied in Coastal Acid Rain Areas

**DOI:** 10.3390/ma17030752

**Published:** 2024-02-04

**Authors:** Junfeng Wang, Shaoxuan Zhang, Qionglin Fu, Yang Hu, Liulei Lu, Zhihao Wang

**Affiliations:** 1College of Civil and Transportation Engineering, Shenzhen University, Shenzhen 518060, China; 2Key Laboratory of Ministry of Education for Tropical Marine Biological Resources Utilization and Protection, College of Fisheries and Life Science, Hainan Tropical Ocean University, Sanya 572022, China; 3School of Civil Engineering and Architecture, Hainan University, Haikou 570228, China

**Keywords:** coastal acid rain environment, orthogonal experiment, durability, chloride ion permeability coefficient, corrosion resistance coefficient

## Abstract

Investigations regarding the preparation and durability of cement-based materials applied in specific coastal acid rain environments are scarce, particularly those involving the addition of four auxiliary cementitious materials (ACMs) to cement for modification. To improve the durability of concrete structures in coastal acid rain areas, a systematic study was conducted regarding the preparation of high-strength and corrosion-resistant cement-based materials using ACM systems composed of fly ash (FA), granulated blast furnace slag (GBFS), silica fume (SF), and desulfurization gypsum (DG) instead of partial cement. Through an orthogonal experimental design, the effect of the water–binder ratio, cementitious ratio, and replacement cement ratio on the compressive strength, corrosion resistance coefficient, and chloride ion permeability coefficient of the materials were analyzed and the mix proportions of the materials were evaluated and optimized using the comprehensive scoring method. The results show that implementing a FA:GBFS:SF:DG ratio of 2:6:1:1 to replace 60% of cement allows the consumption of calcium hydroxide crystals generated through cement hydration, promotes the formation of ettringite, optimizes the pore structures of cementitious materials, and improves the compressive strength, acid corrosion resistance, and chloride ion permeability of the materials. This study provides a reference for selecting concrete materials for buildings in coastal acid rain environments.

## 1. Introduction

The deterioration in the durability of concrete structures in coastal environments is primarily attributed to the corrosion and strength degradation of steel bars in concrete caused by chloride ion penetration [1]. Under certain climatic conditions, such as in concrete structures in the central and eastern regions of China, as well as in the Pearl River Delta region of China, which are all located in coastal and acid rain areas, concrete structures are subject not only to the infiltration of chloride in coastal and atmospheric environments [2,3] but also to the erosion of H+, sulfate, and nitrate in acid rain [4]. Therefore, the durability and service lives of buildings in coastal acid rain areas are threatened by acid corrosion and chloride erosion [5]. Many scholars [6,7,8,9,10,11] have investigated the effect of single environments, such as chloride or acidic environments, on the durability of cement-based materials; however, methods to improve the performance of cement-based materials through various auxiliary cementitious materials (ACMs) in complex coastal acid rain environments are rarely investigated.

Research has shown that when an ACM, such as fly ash (FA) [12,13,14], granulated blast furnace slag (GBFS) [12], or silica fume (SF) [15], is added to cement, the slow pozzolanic effect reduces the strength of cement-based materials during the early hydration period, although their strength improves in the later hydration period. The proportion of cement replaced by mineral admixtures in the above studies was mostly 5–20%. This was because the incorporation of a large amount of GBFS would not only lead to a significant decrease in the compressive strength of the cementitious material system but also lead to an increase in the microstructure porosity of the system [16,17,18]. The high content of FA would lead to the deterioration of the fracture toughness of the cementitious material system, forming a brittle material [19]. The addition of two or three ACMs to cement imposes a better effect as compared with their sole addition. Additionally, compared with the sole addition of FA or SF, the addition of FA and SF composite cementitious materials to cement can improve the compressive strength, chloride ion penetration resistance, and acid corrosion resistance of cement-based materials [20,21,22,23]. Compared with adding GBFS alone, adding desulfurization gypsum (DG) and GBFS composite cementitious materials to cement can improve the chloride ion penetration resistance of cement-based materials [24,25]. Meanwhile, compared with single or double addition, adding FA, GBFS, and SF composite cementitious materials to cement has been reported to increase the compressive strength of cement-based materials [26]. Therefore, the addition of three or fewer ACMs has been investigated to improve the strength and durability of cement-based materials. However, studies regarding methods to improve the performance of cement-based materials by adding four or more ACMs to cement, particularly when applied in coastal acid rain areas, are rarely performed. This is because the corrosion of acidic substances and chloride ion penetration in these particular areas affect the durability and service lives of concrete structures, which must be considered deliberately [5].

In this study, a FA–GBFS–SF–DG ACM system was incorporated into cement via an orthogonal experimental design to investigate the effect of the water–binder ratio, cementitious ratio, and replacement cement ratio on the compressive strength, acid medium erosion resistance, and chloride ion penetration resistance of cement-based materials. An appropriate mix proportion of a cement-based material with good durability and high strength was selected, thus providing a reference for selecting concrete materials for buildings in coastal acid rain environments.

## 2. Materials and Methods

### 2.1. Raw Materials

In this study, FA, GBFS, SFSF and DG were selected to replace part of ordinary cement to form a composite cementitious material system. The reason why FA, GBFS, SF and DG were selected as ACMs was that FA [27,28], GBFS [28], and SF [29] affect the hydration and microstructures of cementitious materials through their filling effect and pozzolanic reaction, thus effectively improving the performance of composite cementitious materials. The addition of DG [30] helps increase the number of ettringite (AFt) crystals and promote the development of strength.

P·II 52.5 grade cement was purchased from China Huarun Cement Holdings Co., Ltd. (Shenzhen, China); FA, GBFS, and DG were purchased from China Baowu Environmental Science Shanxi Resource Recycling Co., Ltd. (Shanghai, China); and SF was purchased from China Aiken Silicone Co., Ltd. (Shanghai, China). The chemical compositions, microstructures, and particle size distributions of the raw materials were measured using a ZSX Primus II X-ray fluorescence spectrometer (Lixue, Beijing, China), a scanning electron microscope (GeminiSEM300-71-11, ZEISS, Oberkochen, Germany), and a laser particle analyzer (Mastersizer 3000, Malvern, UK), respectively. The results are shown in Table 1 and Figure 1 and Figure 2. The FA selected comprised spherical silicon aluminum oxide particles with an average particle size of 19.2 µm [31]. The SF with an average particle size of 18.4 µm comprised clusters formed by the interconnection of fine spheres of silica [32]. The main chemical components of the GBFS with an average particle size of 10.5 µm were calcium oxide, silicon dioxide, and aluminum oxide, and its morphology was mainly irregular with sharp edges and corners, which was due to different grinding technologies used [33]. The DG comprised non-uniform-sized calcium sulfate rod-shaped particles with an average particle size of 33.1 µm [34]. Tap water was used as the test water, and Chinese International Organization for Standardization (ISO) standard sand was used.

### 2.2. Methods

#### 2.2.1. Mix Proportion Optimization Design

To investigate the effects of different factors on the mechanical properties and durability of the mortars, orthogonal experiments were conducted based on three influencing factors: A (water–binder ratio), B (cementitious ratio) (mass ratio of FA, GBFS, SF, and DG), and C (replacement cement ratio). The experimental parameters and their values are listed in Table 2.

Based on the factors and levels presented in Table 2, an orthogonal experimental design table was constructed using the IBM SPSS Statistics 19 software, as shown in Table 3; the mortar mix proportions obtained from Table 3 are listed in Table 4. The effects of different factors on the compressive strength and durability of the mortars were investigated by controlling the cement–sand ratio to 1:3. The mortars were prepared and cured according to the GB/T 17671-2021 Chinese standard [35].

#### 2.2.2. Mix Proportion Optimization Design

The compressive strength values of the mortars after 28 and 90 days of curing were determined in accordance with the GB/T 17671-2021 Chinese standard [35].

#### 2.2.3. Durability Test

Acid corrosion resistance test

Most of the acid rain in China is of the sulfuric acid type (rainwater with a pH value of less than 5.6). To conduct accelerated corrosion tests, sulfuric acid and nitric acid solutions (with a molar ratio of 9:1) were mixed to form an acidic corrosion solution with pH = 1. Subsequently, the 28-day specimens were immersed in an acidic corrosion solution at room temperature (approximately 25 °C), and a compressive strength test was performed on those specimens when the surface of the specimens peeled off (approximately 90 days). To compare the corrosion resistance of the specimens immersed in the acidic corrosive solution, the corrosion resistance coefficient was evaluated based on the GB/T 50082-2009 Chinese standard [36], and the calculation is shown in Equation (1) [36]. Because the pH value of the solution changed with the immersion time, it was replaced with a fresh acidic corrosion solution when the pH value of the corrosion solution increased to approximately 5 after two weeks of immersion.
(1)$$K_{f}=\frac{f_{c}}{f_{c0}}\times 100\%$$Here, *Kf* is the corrosion resistance coefficient (%), *f_c_* is the compressive strength of the specimen immersed in an acidic corrosion solution (0.1 MPa), and *f_c_*_0_ is the compressive strength of the specimen immersed in clean water (0.1 MPa).

2.Chloride ion permeability coefficient test

The chloride ion permeability coefficient was measured using the Nernst–Einstein Lab (NEL) method based on ion diffusion and electromigration [37]. Specimens measuring 100 mm × 100 mm × 50 mm were prepared based on the mix proportion listed in Table 4. After 28 days of standard curing, the specimens were placed in a vacuum water-retaining device and vacuum salt-retained for 24 h with a 4 mol/L NaCl solution. The chloride ion permeability coefficient was measured using a chloride diffusion coefficient tester (NELD-CCM550, NELD, Beijing, China) at 1–10 V, and the results were recorded after 15 min. The solution in the instrument contained 4 mol/L NaCl solution.

#### 2.2.4. X-ray Diffraction (XRD) Analysis

The cement paste cured for 28 days was ground, and alcohol was added to it during the grinding process to reduce the effect of carbon dioxide and moisture in air on the sample. The ground sample was dried at 45 °C and then sieved through a 0.15 mm sieve for analysis via X-ray diffraction (D8 Advance, Bruker, Leipzig, Germany). The instrument was operated at a voltage and current of 40 kV and 40 mA, respectively, and scanned in intervals of 0.02° in the 2θ range of 5–70°.

#### 2.2.5. Microscopic Morphology Observation

After the cement paste was cured for 28 days, the sample was crushed and soaked in alcohol and then baked at 45 °C for 5 days. Subsequently, it was fixed onto the loading platform with a conductive adhesive for gold spraying treatment. The microstructures of the samples were observed using a scanning electron microscope (SEM, GeminiSEM300-71-11, ZEISS, Germany) at a voltage of 3 kV.

#### 2.2.6. Pore Structure Test

The 28 days hardened cement paste was crushed, sieved to obtain particles with a size of 1–5 mm, soaked in alcohol for 5 d, dried at 45 °C, and stored under vacuum. The dried samples were tested using a mercury porosimeter (AutoPore V9600, Micromeritics, Norcross, GA, USA) in the pressure range of 0–60,000 psi.

#### 2.2.7. Statistical Analysis

The corrosion resistance coefficient and chloride ion permeability coefficient of the sample after 28 d of curing, as well as the compressive strength of the sample after 90 days of curing, were assessed via statistical analysis. When 60% of cement is replaced by ACMs, the pozzolanic effect of these materials in samples cured for 28 days will be misjudged; therefore, the compressive strength of the samples cured for 90 days (post hydration) was selected for analysis.

1Range analysis

The ranges (R values) of the three influencing factors (A, B, and C) were calculated based on the experimental results (compressive strength, chloride ion permeability coefficient, and corrosion resistance coefficient). Rj represents the range of factors in the jth column, which is the difference between the maximum and minimum indicator values at each level under the jth column factor; Kji represents the sum of the experimental results for the corresponding jth factor at the ith level; and kji is the average value of Kji [38].

2Variance analysis

The IBM SPSS statistical 22 software was used to perform variance analysis on the factors (independent variables) and their experimental results (dependent variables). Based on the R value in the range analysis, the factor with the least effect on the results was selected as the blank control. The independent variables were factors A, B, and C. The dependent variables included the 90 days compressive strength, corrosion resistance coefficient, and chloride ion permeability coefficient. By calculating the *p* value from variance analysis, one can determine whether a significant difference exists between two variables (see [39,40]). The effect of a factor is significant when 0.01 < *p* < 0.05 whereas it is extremely to be significant when *p* ≤ 0.01 [41].

3.Comprehensive scoring evaluation

Comprehensive scoring evaluation is established in the context of complex data by assigning corresponding weights to multiple indicators and transforming them into a comprehensive indicator (*CI*) for the overall evaluation of a certain feature. The *CI* is calculated as shown in Equation (2) [42].
(2)$$CI={\left(d_{1}d_{2}\ldots d_{k}\right)}^{\frac{1}{k}}$$Here, *k* is the number of indicators and di is the standard value.

Individual data from different indicator results typically cannot be compared directly; in fact, they require conversion into data of the same specification and scale before comparison can be performed. This transformation typically involves assigning weight coefficients and standardized values to indicator data, i.e., performing 0–1 standardization calculation on the indicator data to obtain a variable pertaining to the ‘standardized value’, which is represented by *d_i_*. This can be calculated as shown in Equations (3) and (4) [43].

The higher the values of the indicator data calculated below are, the better the results are.
(3)$$d_{i}=\frac{Y_{i}-Y_{min}}{Y_{max}-Y_{min}}$$

Meanwhile, the lower the values of the indicator data calculated below are, the better the results are.
(4)$$d_{i}=\frac{Y_{max}-Y_{i}}{Y_{max}-Y_{min}}$$Here, *Y_max_* and *Y_min_* are the maximum and minimum values for each indicator, respectively.

## 3. Results and Analysis

### 3.1. Compressive Strength

Figure 3 shows the compressive strength of the samples. Compared with the 28 days samples, the 90 days samples demonstrated a higher compressive strength, thus indicating that an increase in age is conducive to the development of cement hydration and thus improves the compressive strength [44]. Under 90 days of curing, the compressive strength of sample Z3 was the highest, i.e., 73.1 MPa, followed by 70.9 MPa in sample Z9, which were 9.1% and 5.8% higher than that of the control sample (Z6), respectively. This indicates that the addition of an appropriate amount of the FA–GBFS–SF–DG ACM system to cement can improve the compressive strength, which is mainly due to the filling effect of ACMs and the pozzolanic effect of calcium hydroxide (CH) generated during cement hydration to form a dense hydrated calcium silicate (C-S-H) gel [45].

### 3.2. Durability 

#### 3.2.1. Acid Corrosion Resistance

Figure 4 shows the corrosion resistance coefficients of the samples. Among the samples, sample Z7 indicated the highest corrosion resistance coefficient, i.e., 97%, which was 6% higher than that of the control sample (Z1). This indicates that the addition of a reasonable FA–GBFS–SF–DG ACM system to cement can effectively improve the acid corrosion resistance of cementitious materials, which is attributed to the lower CH content in the sample compared with that in the control sample. The ACM replaces some of the cement to reduce the amount of CH generated via cement hydration; meanwhile, the CH crystals are consumed by the pozzolanic effect, which reduces the chemical reaction between the CH crystals and acid substances in cement-based materials [46,47].

#### 3.2.2. Resistance to Chloride Ion Penetration

Figure 5 shows the chloride ion permeability coefficients of the 28 days samples. Among the samples, sample Z9 exhibited the best chloride ion permeability resistance, and its chloride ion permeability coefficient was 2.0 × 10^−13^ m2/s, which was 89% lower than that of the control sample (Z6), thus indicating that the addition of the FA–GBFS–SF–DG ACM system in cement can significantly improve the chloride ion permeability resistance of the sample, which may be attributed to the microaggregate effect and pozzolanic effect of the ACM system to promote the compactness of its cement-based material structures [48].

### 3.3. XRD

Figure 6 shows the XRD pattern of the 28 days cement paste. As shown, compared with the case of the reference sample (Z1), the peak value of the CH crystal (observed at 18.1°, 29.4°, and 34.1°) in sample Z7 was lower and the peak of the ettringite (AFt) crystal (9.1° and 15.8°) was higher. Compared with the case of the control sample (Z6), the peak value of CH crystals in the Z3 and Z9 samples decreased whereas the peak value of the AFt crystals increased, thus indicating that the addition of the FA–GBFS–SF–DG ACM systems in cement can significantly reduce the content of CH crystal and increase the AFt crystal content in cement pastes [49].

### 3.4. Microscopic Morphology Observation

Figure 7 shows the scanning electron microscopy (SEM) images of the 28 days cement paste. Unhydrated FA and SF spherical particles were observed in the Z3, Z7, and Z9 samples, but not in the control samples (Z6 and Z1), thus indicating that the raw materials had not been fully hydrated and that their pozzolanic effect had not been fully exerted in the cement paste after curing for 28 days [50].

### 3.5. Pore Structure

Figure 8 shows the pore size distribution and pore volume fractions of the samples after 28 days of curing. As shown in Figure 8a, the pore peak values of the Z3 and Z9 samples were 62.55 and 50.39 nm, respectively whereas the pore peak value of the control sample (Z6) was 65.65 nm; the pore peak value of Z7 sample was 64.23 nm whereas that of the control sample (Z1) was 68.06 nm. This indicates that the addition of the FA–GBFS–SF–DG ACM systems to the cement can refine the pore structures of the samples. In addition, the pore peak value of sample Z9 was the lowest, followed by that of sample Z7. According to Wu [51], pore sizes in cement paste can be classified into four categories: harmless pores (<20 nm), less harmful pores (20–50 nm), harmful pores (50–200 nm), and more harmful pores (>200 nm). As shown in Figure 8b, the volume fractions of harmless and less harmful pores in the Z9 sample were the highest among the samples, followed by those of the Z7 sample, thus indicating that the pore structure of sample Z9 was the densest among the samples, followed by that of sample Z7.

### 3.6. Statistical Analysis

#### 3.6.1. Range Analysis

The results of the range analysis results are presented in Table 5. As shown, factor C (replacement cement ratio) had a prominent effect on the compressive strength and chloride ion permeability coefficient of the samples whereas factor A (water–binder ratio) had a significant effect on the corrosion resistance coefficient of the material compared with the other two factors (B and C); meanwhile, factor B (cementitious ratio) had a less prominent effect on the performance of the samples. The effects of these factors on the indicator results can be elucidated from the mean kji values listed in Table 5. As the replacement cement ratio increased, the chloride ion penetration resistance of the samples increased, and their compressive strength first increased and then decreased; meanwhile, as the water–binder ratio increased, the corrosion resistance coefficient of the samples first increased and then decreased, and their compressive strength decreased.

#### 3.6.2. Variance Analysis

To verify the significance of the factors affecting the performance based on the range analysis results (Table 5), a variance analysis was conducted on the performance results, which are shown in Table 6. As shown, the *p* values of the corresponding factors, i.e., the compressive strength and corrosion resistance coefficient, exceeded 0.05, thus indicating that the effect of these factors was not significant. Therefore, based on the range analysis (Table 5), although factors C and A affected the compressive strength and corrosion resistance coefficient, respectively, the effects were insignificant, and the *p* value of the effect of factor C on the chloride ion permeability coefficient was less than 0.01, thus indicating that factor C had a significant effect on the chloride ion permeability coefficient. This is consistent with the effect of factor C based on the range analysis (Table 5), thus indicating that the replacement cement ratio significantly affects the resistance to chloride ion penetration.

#### 3.6.3. Comprehensive Scoring Evaluation

Table 7 presents the CI values of the samples. As shown, among the samples, sample Z9 indicated the highest OD value, followed by sample Z2. This indicates that compared with various other ACMs, when a FA:GBFS:SF:DG ratio of 2:6:1:1 was implemented, the FA–GBFS–SF–DG system effectively improved the compressive strength, chloride ion permeability, and acid corrosion resistance of the cement-based materials by replacing 30% or 60% cement.

## 4. Discussion

Our results indicate that the addition of a reasonable proportion of the FA–GBFS–SF–DG ACM system to cement can improve the compressive strength (Figure 3), acid corrosion resistance (Figure 4), and resistance to chloride ion penetration (Figure 5) of cementitious materials. Sample Z3 indicated the highest compressive strength (Figure 3) whereas sample Z7 indicated the highest acid corrosion resistance (Figure 4), thus demonstrating that samples Z3 and Z7 offer strong advantages in a single performance. However, environments in coastal acid rain areas containing various corrosive media such as chloride ions and acidic substances may severely degrade the service lives of cement-based materials. By contrast, the Z9 sample indicated the lowest chloride ion permeability coefficient, densest pore structure, and highest CI value, along with both high strength and corrosion resistance, thus demonstrating it potential applications in coastal acid rain areas.

The high compressive strength and good durability of the Z9 sample are related to its low water–binder ratio. A lower water–binder ratio will increase the adhesion and compactness between the slurry and the aggregate, reduce the porosity (Figure 8), and form a denser microstructure [52,53]. Therefore, the decrease in the water–binder ratio is one of the reasons for improving the properties of cementitious materials. In addition to the influence of a low water–binder ratio, the superior performance of sample Z9 was attributed to the microaggregate and pozzolanic effects of the FA–GBFS–SF–DG ACM system in the cement, as shown in Figure 9. As shown in the figure, gaps were present between the cement particles in ordinary mortar, and a reasonable FA–GBFS–SF–DG system can exert a microaggregate effect in mortar mixed with various ACMs. When a FA:GBFS:SF:DG = 2:6:1:1 ratio was implemented to replace 60% of cement, the FA, GBFS, SF, and DG in cement evenly occupied the voids between cement particles, improved the compactness of its cement-based material, and refined its pore structures [54]. Therefore, the volume fraction of harmless and less harmful pores in the 28 days Z9 sample was higher than those of the other samples, and its pore structures was more dense, which could effectively reduce the permeability of chloride ions into the concrete and improve its compressive strength and durability [55].

The main products of cement hydration in ordinary mortar are CH crystals, C-S-H gel, hydrated calcium aluminates (C-A-S-H), and tetracalcium aluminate [56]. However, the FA, GBFS, and SF added to the cement contained significant amounts of active SiO_2_ and Al_2_O_3_. These active ingredients can react with the CH crystals generated by cement hydration in the presence of water, thus consuming CH crystals in the Z9 sample (Figure 6) and generating C-S-H and C-A-S-H gels [57,58]. This not only increases the compressive strength of the samples but also improves their acid corrosion resistance [59,60]. Meanwhile, DG can be used as an activator of FA and GBFS to enhance their pozzolanic effects [61]; however, it can react both with alumina in FA and GBFS as well as with C-A-S-H crystals in cement pastes to promote the formation of AFt crystals (Figure 6) [62]. The hydration products of C-S-H and C-A-S-H gels can increase the physical adsorption capacity of chloride ions [63]. In addition to physical adsorption, because FA and GBFS contain large amounts of highly active Al_2_O_3_, a large number of Al ions lead to the formation of calcium aluminate monosulfate (AFm) by tricalcium aluminate, and AFm forms Friedel’s salt with chloride, which increases the chemical adsorption capacity of chloride ions in cementitious materials [64,65,66]. The pozzolanic reaction of FA and GBFS not only promotes the formation of C-S-H and C-A-S-H gels to increase the physical adsorption of chloride ions but also promotes the formation of Friedel’s salt to increase the chemical adsorption of chloride ions. Therefore, compared with other samples, the Z9 sample has a lower chloride ion permeability coefficient (Figure 5).

Although the compressive strength of the Z9 sample after 28 days of curing was lower than that of the control sample (Z6), its 90 days compressive strength exceeded 70 MPa, which was 5% higher than that of the control sample (Z6) (Figure 3). This was because some ACMs in the samples had not been fully hydrated after curing for 28 days and the pozzolanic effect had not yet been fully realized [50]. Therefore, spherical particles of FA and SF were observed in the Z9 sample after curing for 28 days (Figure 7). 

It can be seen from the above analysis that the Z9 sample has excellent durability and high compressive strength. However, in the practical engineering application of cementitious materials, the cost and carbon emissions of materials should also be considered. In terms of cost, the annual average price of each raw material per ton is listed in Table 8, and the data come from www.mysteel.com (accessed on 17 January 2024) in China. From Table 8, it can be seen that for the Z9 sample, the cost per ton of cementitious material is 515 RMB. When the cement replacement rate is 60%, the cost per ton of its cementitious material is 329–349 RMB, decreased by 32–36%. In terms of carbon emissions, carbon dioxide emissions are mainly related to cement production. Adding auxiliary cementitious materials to cement can reduce the amount of cement used, which is an effective measure to reduce carbon dioxide emissions [67]. According to statistics, the global cement production in 2020 was 4.1 × 1012 kg [68], expected to reach approximately 6.0 × 10^12^ kg by 2050 [69]. Since its production emits 0.5–0.7 kg CO_2_ per kg of cement [70], it is expected to emit 3.0–4.2 × 10^12^ kg of CO_2_ by 2050. The application of Z9 samples with ACMs replacing 60% cement will reduce CO_2_ emissions by 60%, which is of positive significance for achieving the low-carbon emission reduction target by 2050. Therefore, considering the material properties, economic benefits, and environmental benefits comprehensively, the development of Z9 samples will provide important technical references for the selection of concrete structural materials in coastal acid rain areas.

The research results indicate that using a FA:GBFS:SF:DG ratio of 2:6:1:1 instead of 60% cement can improve the compressive strength, acid corrosion resistance, and chloride ion permeability of the materials. However, this study was conducted on mortar with aggregate particles smaller than 4.75 mm, and further research is needed on concrete with aggregate particles larger than 4.75 mm.

## 5. Conclusions

In this study, the effects of adding a FA–GBFS–SF–DG system to cement on the compressive strength, corrosion resistance coefficient, and chloride ion permeability coefficient of cement-based materials were investigated. The mix ratio of the cement-based materials was optimized via an orthogonal experiment and a comprehensive scoring method. The conclusions obtained were as follows: (1)The addition of the FA–GBFS–SF–DG cementitious system to cement can increase the compressive strength of the cementitious material by up to 9.1% and the acid corrosion resistance coefficient by 6%, and the chloride ion permeability coefficient is reduced by 89% compared to ordinary cement materials.(2)The addition of the FA–GBFS–SF–DG cementitious system to cement consumed CH crystals produced via cement hydration, promoted the formation of AFt, and optimized the pore structure of the cementitious materials.(3)Compared with the water–binder and cementitious material ratios, the cement replacement ratio significantly reduced the chloride ion permeability coefficient of the cementitious materials.(4)When a FA:GBFS:SF:DG ratio of 2:6:1:1 was implemented to replace 60% of cement, the compressive strength, acid corrosion resistance, and resistance to chloride ion permeability improved. This ratio is expected to benefit concrete structures in coastal acid rain environments.

## Figures and Tables

**Figure 1 materials-17-00752-f001:**
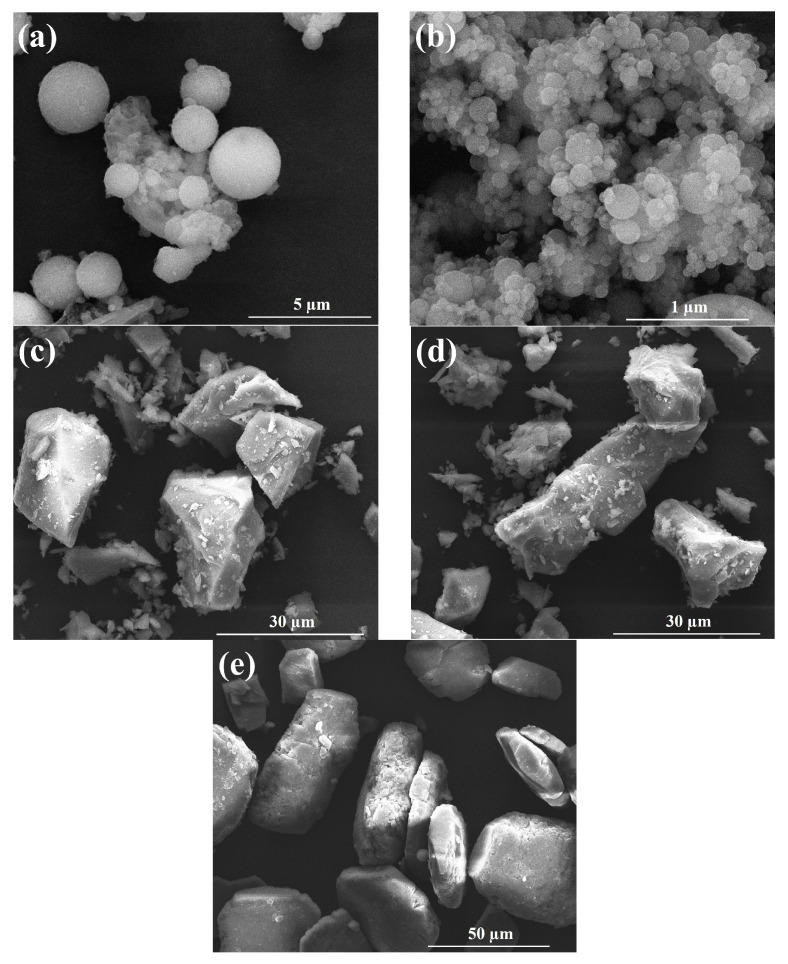
Micromorphology of raw materials: (**a**) fly ash (FA); (**b**) silica ash (SF); (**c**) granulated blast furnace slag (GBFS); (**d**) cement; (**e**) desulfurization gypsum (DG).

**Figure 2 materials-17-00752-f002:**
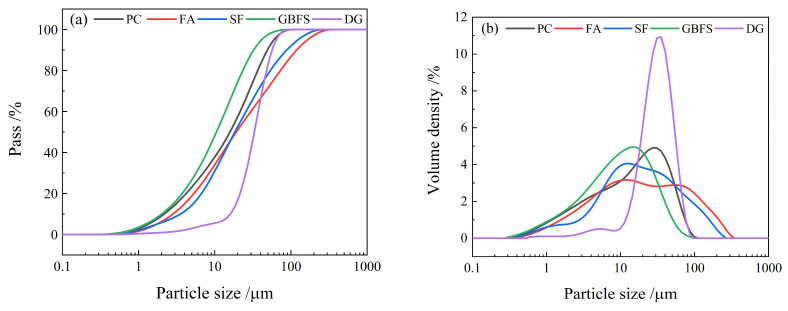
Cumulative distribution (**a**) of raw material particles; interval distribution (**b**) of raw material particles.

**Figure 3 materials-17-00752-f003:**
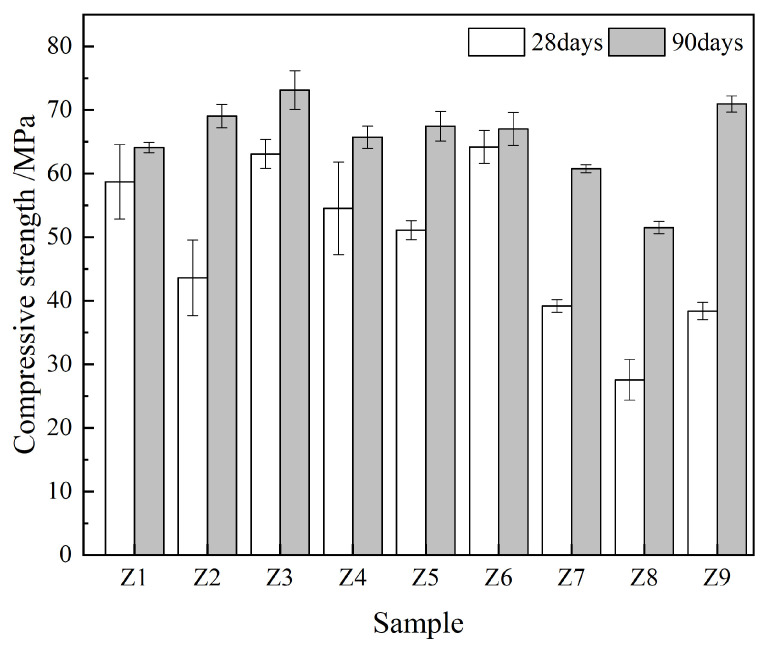
Compressive strength of the samples.

**Figure 4 materials-17-00752-f004:**
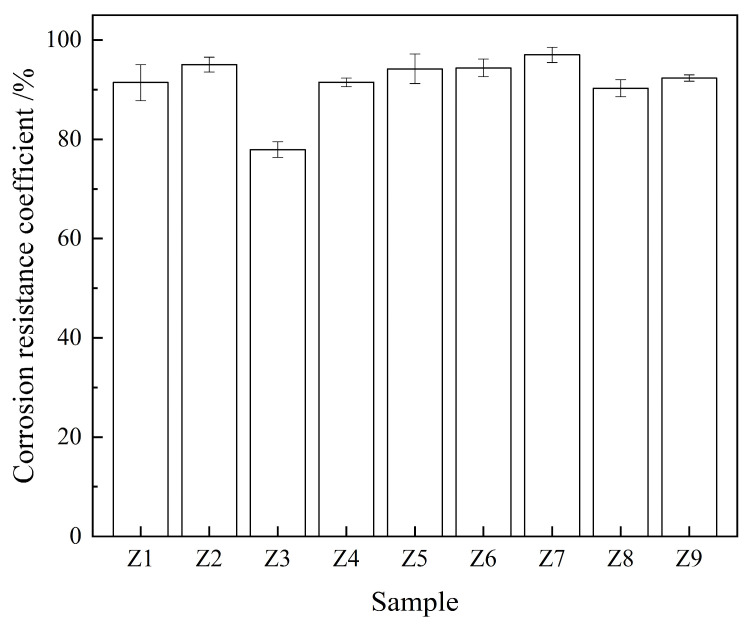
Corrosion resistance coefficient of the samples.

**Figure 5 materials-17-00752-f005:**
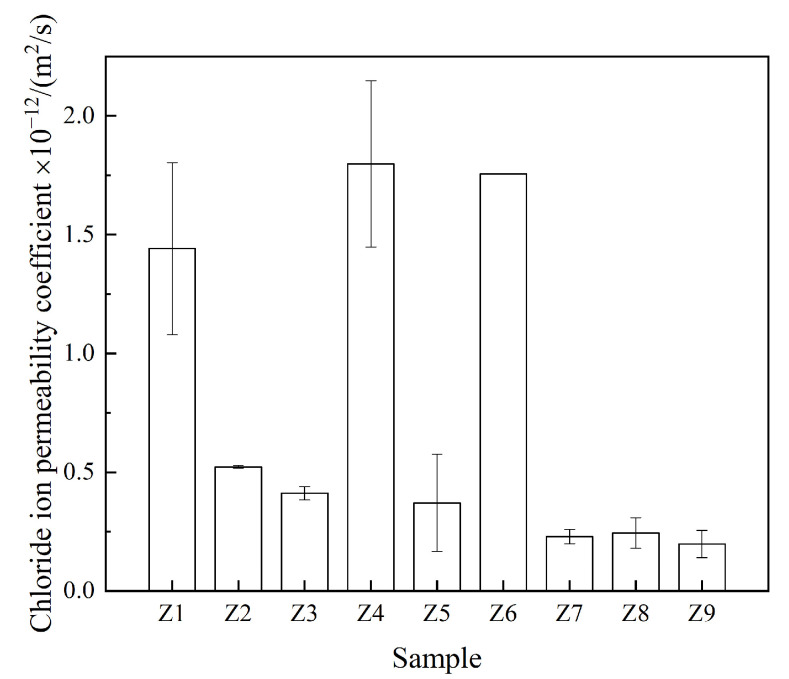
Chloride ion permeability coefficient of the 28 days samples.

**Figure 6 materials-17-00752-f006:**
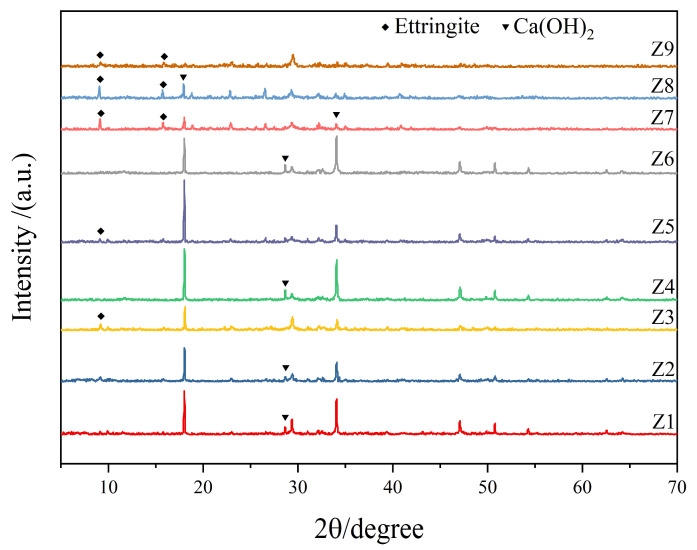
XRD pattern of the 28 days cement pastes.

**Figure 7 materials-17-00752-f007:**
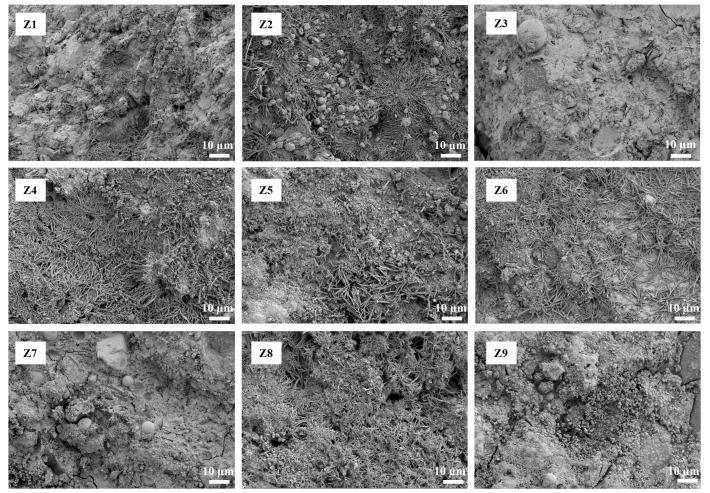
SEM images of the 28 days cement pastes.

**Figure 8 materials-17-00752-f008:**
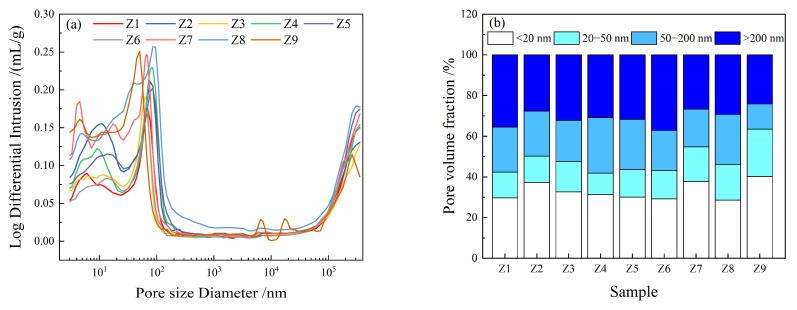
Pore size distribution (**a**) of the samples; pore volume fraction (**b**) of the samples.

**Figure 9 materials-17-00752-f009:**
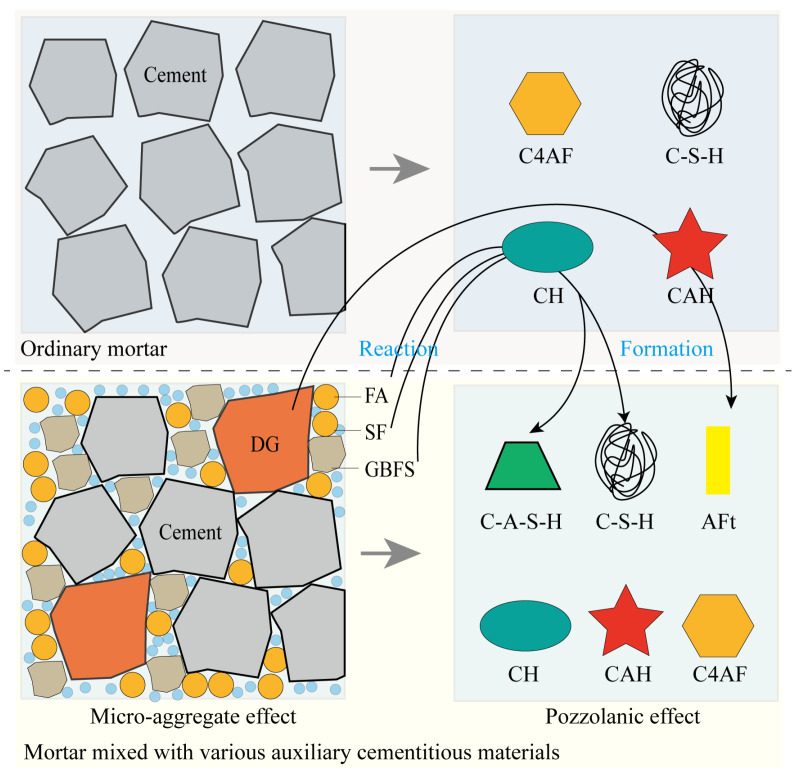
Schematic diagram of filling and pozzolanic effects of FA-GBFS-FA-DG ACM system in cement.

**Table 1 materials-17-00752-t001:** Chemical composition of raw materials used in current study.

Materials	CaO	SiO_2_	Fe_2_O_3_	Al_2_O_3_	SO_3_	K_2_O	TiO_2_	MgO	N
Cement	72.0	15.3	3.56	3.45	2.93	0.79	0.19	0.55	0.76
FA	11.2	41.8	4.01	36.6	1.60	0.76	1.28	0.72	0.52
GBFS	48.6	28.0	0.48	14.2	1.98	0.28	0.74	4.91	0.00
SF	1.44	96.6	0.14	0.49	0.09	0.39	0.00	0.18	0.26
DG	47.4	2.48	0.52	1.38	47.5	0.12	0.08	0.22	0.00

Note: FA—fly ash; GBFS—granulated blast furnace slag; SF—silica fume; DG—desulfurization gypsum.

**Table 2 materials-17-00752-t002:** Factors and levels of orthogonal experimental design.

Levels	Experimental Factors		
	A (Water–Binder Ratio)	B (Cementitious Ratio)	C (Replacement Cement Ratio)
1	0.55	4:4:1:1	0%
2	0.45	6:2:1:1	30%
3	0.50	2:6:1:1	60%

**Table 3 materials-17-00752-t003:** Orthogonal experiment design.

Mix NO.	Factors		
	A	B	C
Z1	3 (0.50)	2(6:2:1:1)	1 (0.0%)
Z2	1 (0.55)	2	2 (30%)
Z3	2 (0.45)	1(4:4:1:1)	2
Z4	1	1	1
Z5	3	3(2:6:1:1)	2
Z6	2	3	1
Z7	3	1	3 (60%)
Z8	1	3	3
Z9	2	2	3

**Table 4 materials-17-00752-t004:** Mix proportions of mortars.

Mix NO.	Quality (g)						
	GBFS	FA	SF	DG	Cement	Sand	Water
Z1	0	0	0	0	450	1350	225.0
Z2	81	27	13.5	13.5	315	1350	247.5
Z3	54	54	13.5	13.5	315	1350	202.5
Z4	0	0	0	0	450	1350	247.5
Z5	27	81	13.5	13.5	315	1350	225.0
Z6	0	0	0	0	450	1350	202.5
Z7	108	108	27	27	180	1350	225.0
Z8	54	162	27	27	180	1350	247.5
Z9	162	54	27	27	180	1350	202.5

**Table 5 materials-17-00752-t005:** Range analysis results.

Indicator Value	Compressive Strength/MPa			Corrosion Resistance Coefficient/%			Chloride Ion Permeability Coefficient /(m^2^/s)		
	A	B	C	A	B	C	A	B	C
K_j1_	186	200	197	277	266	277	2.57 × 10^−12^	2.44 × 10^−12^	4.99 × 10^−12^
K_j2_	190	204	210	264	279	268	2.37 × 10^−12^	2.16 × 10^−12^	1.40 × 10^−12^
K_j3_	192	186	183	284	280	280	2.14 × 10^−12^	2.47 × 10^−12^	0.67 × 10^−12^
k_j1_	62.0	66.5	65.6	92.4	88.8	92.3	0.86 × 10^−12^	0.81 × 10^−12^	1.67 × 10^−12^
k_j2_	63.4	68.0	69.8	88.1	93.0	89.4	0.79 × 10^−12^	0.72 × 10^−12^	0.47 × 10^−12^
k_j3_	64.1	62.0	61.1	94.5	93.3	93.4	0.71 × 10^−12^	0.82 × 10^−12^	0.22 × 10^−12^
R	2.10	6.00	8.70	6.40	4.50	4.00	0.15 × 10^−12^	0.10 × 10^−12^	1.45 × 10^−12^
Ranking of impact	C > B > A			A > B > C			C > A > B		

**Table 6 materials-17-00752-t006:** Variance Analysis results.

Dependent Variable	Independent Variable (Factor)	Type III Sum of Squares	Df (Free Degree)	F Value	*p* Value
Compressive strength/MPa	B	59.4	2	0.79	0.51
	C	115	2	1.53	0.32
	Error	150	4		
	R^2^ = 0.54 (Adjust R^2^ = 0.08)				
Corrosion resistance coefficient/%	A	63.3	2	0.869	0.49
	B	36.8	2	0.506	0.64
	Error	145	4		
	R^2^ = 0.41 (Adjust R^2^ = 0.19)				
Chloride ion permeability coefficient/(m^2^/s)	A	0.03 × 10^−12^	2	1.01	0.44
	C	3.57 × 10^−12^	2	128	0.00
	Error	0.06 × 10^−12^	4		
	R^2^ = 0.99 (Adjust R^2^ = 0.97)				

**Table 7 materials-17-00752-t007:** CI values of the samples.

Mix NO.	Compressive Strength/MPa	Corrosion Resistance Coefficient/%	Chloride Ion Permeability Coefficient/(m^2^/s)	CI Values
Z1	64.1	91.4	1.44 × 10^−12^	0.45
Z2	69.0	95.1	0.52 × 10^−12^	0.83
Z3	73.1	78.0	0.42 × 10^−12^	0.00
Z4	65.7	91.5	1.80 × 10^−12^	0.00
Z5	67.4	95.1	0.48 × 10^−12^	0.82
Z6	67.0	94.0	1.76 × 10^−12^	0.25
Z7	60.8	97.0	0.23 × 10^−12^	0.75
Z8	51.5	90.7	0.24 × 10^−12^	0.00
Z9	70.9	92.4	0.20 × 10^−12^	0.87

**Table 8 materials-17-00752-t008:** The average annual price of each raw material.

Materials	Cement	FA	GBFS	SF	DG
Price per ton/RMB	515	130.2	212.5	800	46.3

## Data Availability

The original contributions presented in the study are included in the article, further inquiries can be directed to the corresponding authors.

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
