# Peer review of "Preparation and Performance Study of High-Strength and Corrosion-Resistant Cement-Based Materials Applied in Coastal Acid Rain Areas"

_materials, 2024, doi:10.3390/ma17030752_

Round 1

Reviewer 1 Report

Comments and Suggestions for Authors

Congratulations on a very interesting article on the influence of mineral additives on the strength properties of concrete.

The study examined the effect of adding the FA-GBFS-SF-DG system to cement on the compressive strength, corrosion resistance coefficient and chloride ion permeability coefficient of cement-based materials. The mixing ratio of cement-based materials was optimized using orthogonal experiment and comprehensive scoring method.

Mineral additives replacing cement in the form of FA-GBFS-SF-DG may increase or replace the obtained compressive strength, but it is worth adding the prices of individual mineral additives and the price of the mortar mixture to this analysis, which would make it clear whether it is profitable in the financial calculation.

The possibility of using less cement in concrete is the right direction, especially when we have a common goal of reducing CO2 emissions in the world, so it is worth calculating the carbon footprint of individual recipe compositions and including it in the analysis.

An added value for this type of work would be to perform tests on concrete samples with the addition of aggregate with a grain size > 2 mm.

A valuable element of the work is, of course, the information that the use of the ratio FA:GBFS:SF:DG = 2:6:1:1 to replace 60% of cement improved compressive strength, acid corrosion resistance and resistance to chloride ion permeability. This ratio is expected to be beneficial for concrete structures in coastal acid rain environments.

After completing the indicated elements, please send your work again for verification.

Author Response

Dear Reviewers,

Thank you for your comments concerning our manuscript entitled “Preparation and performance study of high-strength and corrosion-resistant cement-based materials applied in coastal acid rain areas” (Manuscript ID: materials-2818727). These comments are all valuable and very helpful for revising and improving our paper, as well as the important guiding significance to our researches. We have studied comments carefully and have made correction which we hope meet with approval. Revised portion are clearly highlighted in the paper, you can distinguish the changes using “Track Changes” function in Microsoft Word. The responds to the reviewer’s comments are as flowing:

Response to Reviewer #1 as follows:

  1. Response to comment:” Mineral additives replacing cement in the form of FA-GBFS-SF-DG may increase or replace the obtained compressive strength, but it is worth adding the prices of individual mineral additives and the price of the mortar mixture to this analysis, which would make it clear whether it is profitable in the financial calculation.”

Response: Line 386-390, prices of individual mineral additives and the price of the mortar mixture have been added.

  1. Response to comment:” The possibility of using less cement in concrete is the right direction, especially when we have a common goal of reducing CO2 emissions in the world, so it is worth calculating the carbon footprint of individual recipe compositions and including it in the analysis.”

Response: Line 391-399, “According to statistics, the global cement production in 2020 was 4.1 × 1012 kg, expected to reach approximately 6.0 × 1012 kg by 2050. According to its production emits 0.5–0.7 kg CO2 per kg of cement, it is expected to emit 3.0-4.2 × 1012 kg of CO2 by 2050. The application of Z9 samples with ACMs replacing 60% cement will reduce CO2 emissions by 60%” has been added.

  1. Response to comment:” An added value for this type of work would be to perform tests on concrete samples with the addition of aggregate with a grain size > 2 mm.”

Response: Line 404-408, “this study was conducted on mortar with aggregate particles smaller than 4.75 mm, and further research is needed on concrete with aggregate particles larger than 4.75 mm.” has been added.

Reviewer 2 Report

Comments and Suggestions for Authors

The authors achieved an increase in strength and acid resistance and a decrease in permeability to chlorides and explained them exclusively by the influence of mineral additives, but did not pay attention to the possible influence of the water-cement (water-binding) ratio, which was very different in the experimental compositions. It is necessary to supplement the research results with a discussion of the influence of water-cement (water-binding ratio).

The authors obtained a significant decrease in permeability for chlorides. It was necessary to rank the influence on the permeability of the density of the structure of the products of hydration and binding of chlorides according to the extent of their penetration by aluminate phases into Aft or Afm-phases.

It is necessary to adjust the conclusion (1) about the increase in permeability by 89%. Permeability decreases.

Author Response

Dear Reviewers,

Thank you for your comments concerning our manuscript entitled “Preparation and performance study of high-strength and corrosion-resistant cement-based materials applied in coastal acid rain areas” (Manuscript ID: materials-2818727). These comments are all valuable and very helpful for revising and improving our paper, as well as the important guiding significance to our researches. We have studied comments carefully and have made correction which we hope meet with approval. Revised portion are clearly highlighted in the paper, you can distinguish the changes using “Track Changes” function in Microsoft Word. The responds to the reviewer’s comments are as flowing:

Response to Reviewer #2 as follows:

  1. Response to comment:” It is necessary to supplement the research results with a discussion of the influence of water-cement (water-binding ratio).”

Response: Line 336-341, the discussion of the influence of water-cement has been added.

Response to comment:” The authors obtained a significant decrease in permeability for chlorides. It was necessary to rank the influence on the permeability of the density of the structure of the products of hydration and binding of chlorides according to the extent of their penetration by aluminate phases into Aft or Afm-phases.”

Response: Line 363-372, We have added content about “Physical adsorption and chemical adsorption of chloride ions by hydration products”

  1. Response to comment:” It is necessary to adjust the conclusion (1) about the increase in permeability by 89%. Permeability decreases”.

Response: Line 417, Conclusion (1) has been changed to “The chloride ion permeability coefficient is reduced by 89 %.”

Round 2

Reviewer 1 Report

Comments and Suggestions for Authors

I would like to thank the authors for the explanations they sent and, after re-analysis, I accept all the content contained in the article submitted for review.